# No pets allowed: Evidence that prolonged grief disorder can occur following the death of a pet

Philip Hyland 🆔 *

Department of Psychology, Maynooth University, Kildare, Ireland

* Philip.hyland@mu.ie

## Abstract

### Background

Prolonged grief disorder (PGD) is a psychiatric disorder in ICD-11 and DSM-5-TR that can only be diagnosed following the death of a person. Despite considerable evidence that people form strong attachments to their pets, and experience high levels of grief following their death, the current guidelines do not allow PGD to be diagnosed following the death of a pet. This study tested several hypotheses to determine if there is anything unique about grief that follows the death of a person versus grief that follows the death of a pet.

### Methods

A nationally representative sample of adults from the United Kingdom ($N = 975$) provided information about different bereavements, their most distressing bereavement, and ICD-11 PGD symptoms.

### Results

One-third (32.6%) of respondents experienced the death of a beloved pet, and almost all had also experienced the death of a human; 21.0% of these people chose the death of their pet as most distressing. The conditional rate of PGD following the death of a pet was 7.5%, similar to many types of human losses. The relative risk of PGD following pet bereavement was 1.27, and pet loss accounted for 8.1% of all PGD cases in the population, both of which were higher than many types of human losses. Full measurement invariance for PGD symptoms was found between people who reported symptoms for a human bereavement and for a pet bereavement.

### Conclusions

People can experience clinically significant levels of PGD following the death of a pet, and PGD symptoms manifest in the same way regardless of the species of the

**Data availability statement:** The dataset used for this study is publicly available on the Open Science Framework (https://osf.io/hymd4/overview).

**Funding:** The author(s) received no specific funding for this work.

**Competing interests:** The author has declared that no competing interests exist.

deceased. Implications associated with excluding diagnosis following pet bereavement are discussed.

## Introduction

"*There is no fundamental difference between man and animals in their ability to feel pleasure and pain, happiness, and misery.*" – Charles Darwin

One of the greatest gifts that was given to us by Darwin's theory of evolution by natural selection was the understanding that human beings exist within the natural order and that we are related to all other living organisms [1]. Darwin's discoveries meant there was no scientific basis to regard humans as exceptional within the animal kingdom. *Homo sapiens sapiens* is undoubtedly an unusual African ape, but it is an African ape. The discoveries of evolutionary biology in the intervening 166 years have only deepened our understanding of how unremarkable we human beings are; learning that we share about 50% of our DNA with cabbage will do that [2,3]. Nevertheless, there are still some areas of science where one hominin species – of which there have been at least 20 [4] – is treated as unique and distinct from all other species. In psychology and psychiatry, one example of this phenomenon is found in the newly codified diagnosis of Prolonged Grief Disorder (PGD).

After decades of debate [5], PGD was included in the 11th version of the International Classification of Diseases (ICD-11) [6] and the revised fifth edition of the Diagnostic and Statistical Manual of Mental Disorders (DSM-5-TR) [7]. Diagnostic requirements vary slightly across the two manuals, but both define PGD by two 'core' symptoms (longing for the deceased and preoccupation with the deceased) and a set of 'associated' symptoms (e.g., intense emotional pain, feelings of guilt or sorrow, difficulty accepting the death). Bereavement is a necessary requirement for PGD, and both manuals stipulate that it can only develop following the death of a person. ICD-11 states that "*A history of bereavement following the death of a partner, parent, child or other person close to the bereaved is required for diagnosis*" [8, p. 348], while DSM-5-TR requires "*The death, at least 12 months ago, of a person who was close to the bereaved individual (for children and adolescents, at least 6 months ago)*" [7, pp. 322–323]. Thus, it is not possible to be diagnosed with PGD following the death of a non-human companion animal (i.e., a pet).

Only a small proportion of bereaved people develop PGD [9]. Although the causal processes are not yet fully understood, the end of a close attachment relationship is central to the process [10]. The empirical evidence shows that the closer the attachment to the deceased person, the higher the risk of developing PGD. Specifically, risk of developing PGD is highest following the death of a child and the death of a spouse [11–14]. There is considerable evidence that humans form close attachments to their pets [15–19]. A survey of 5,073 Americans found that 97% of pet owners considered their pet to be a member of their family, and 51% considered their pet as much a member of their family as any human member [20]. If one accepts that the loss of a

close attachment relationship is central to the development of PGD, then this evidence means it is reasonable to expect that PGD can occur following the death of a beloved pet. Numerous studies have shown that people who have experienced the death of their pet report high levels of grief, and similar in intensity to that which is experienced after the death of a human loved one [21–23].

It is not clear why the death of a pet was excluded from the bereavement criterion for PGD in ICD-11 [6,8] and DSM-5-TR [7]. It is possible that the controversial nature of the diagnosis meant that the different working groups were reluctant to acknowledge that pet loss can lead to PGD for fear of being viewed as unserious – experimental evidence shows that people view PGD following pet loss as less legitimate than PGD following human loss [24] – or for fear of providing critics with another reason to argue that PGD pathologizes normal reactions to death [25]. Another reason may be that the members of these working groups sincerely believed that there is something uniquely special about human beings' attachments to other human beings. Whatever the reason, it is important to test if people bereaved by the death of a pet can experience disordered grief in the manner it is now described in the psychiatric nomenclature.

In this study, data collected from a representative sample of adults living in the United Kingdom (UK) was used to test several research objectives. The first was to determine what proportion of UK adults had experienced the death of a beloved pet. The second was to determine, among those who had experienced the death of a pet, what proportion had also experienced the death of a person, and – when asked to identify their most distressing bereavement – what proportion selected the death of their pet. The third objective was to calculate conditional rates of PGD based on different types of bereavements, including the loss of a parent, sibling, partner, child, other family member, close friend, and beloved pet. The fourth was to determine the relative risk and population-attributable risk percentage for PGD based on each type of bereavement. Finally, the fifth objective was to assess whether PGD symptoms reported following the death of a beloved pet manifest in the same way, or in a different way, to PGD symptoms reported following the death of a beloved human.

Based on the assumptions (a) that humans form strong attachments to their pets [e.g., 19], (b) that humans can experience high levels grief in response to pet loss [e.g., 23], and (c) that there is nothing exceptional about human beings [e.g., 1, 2], the following hypotheses were formulated. First, a non-trivial proportion of people (e.g., > 10%) who had experienced the death of a pet *and* the death of a human would select the death of their pet as their most distressing bereavement. Second, the conditional rate of PGD in response to the death of a pet would be significantly different from zero (thereby indicating that it is possible to develop PGD following pet loss) and comparable to rates of PGD following many types of human loss. Third, the relative risk and population-attributable risk percentages for PGD following pet loss would be positive and comparable to many types of human loss. And fourth, that PGD symptoms would manifest in the same way between those reporting their symptoms in relation to a pet bereavement and those reporting their symptoms in relation to a human bereavement.

## Materials and methods

### Participants and procedures

Data were collected from 975 adults living in the UK by *Qualtrics* who access participants from online research panels. Quota sampling methods were used to construct a sample that represented the UK adult population by sex, age, nation (England, Wales, Scotland, and Northern Ireland), and annual income (<£20,000, £20,00-£39,999, £40,00-£59,999, £60,000-£79,999, £80,000-£99,999, >£100,000). This non-probability-based sampling method means that the sample is unlikely to be fully representative of the UK adult population, but merely that steps were taken to construct a sample reasonably representative of the population. Nevertheless, samples constructed in this way have been shown to be highly representative of the UK population [26].

These data were collected from 1–27 March 2024, and ethical approval was granted by the Social Research Ethics Committee at Maynooth University (SRESC-2023–37628). The survey included several attention check questions, and

*Qualtrics* uses different quality control measures to ensure valid responses including a CAPTCHA security question to enter the survey, removal of responses from duplicate IP addresses or those hiding their location, and removal of any responses deemed to be too quick or demonstrating suspicious patterns of responding. Participants in this study passed all quality control checks. Sociodemographic details are presented in Table 1. The dataset used for this study is publicly available on the Open Science Framework (https://osf.io/hymd4/overview).

## Measures

**Bereavement.** Participants were given the following instruction; "*Bereavement is a common experience and involves someone close to you dying (e.g., a partner, parent, child, close friend). It has been suggested that losing a pet can be distressing for some people. We would like you tell us if you have ever experienced the death of any of the following:*" A list was presented that included 'a parent', 'a brother or sister', 'a partner or spouse', 'a child', 'any other family member (e.g., aunt, uncle, grandparent, cousin)', 'a close friend', and 'a beloved pet'. Participants indicated if they had ever experienced each loss using a 'Yes' or 'No' scale. Participants were then asked to indicate the bereavement they found to be most distressing (i.e., their index bereavement), and how long ago it occurred.

**ICD-11 PGD.** The International Grief Questionnaire with Clinical Checks (IGQ-CC) [27] is a self-report measure of all diagnostic requirements for ICD-11 PGD. Participants answered all questions based on their index bereavement. The IGQ-CC contains two items measuring the 'core' symptoms of longing for, and preoccupation with, the deceased, and three items measuring the 'associated' symptoms (i.e., intense feelings of anger or guilt about the loss, difficulty accepting the death, and intense feelings sadness or numbness since the loss). Participants rated how bothered they had been over the last week on a five-point Likert scale (0 = Not at all, 1 = A little bit, 2 = Moderately, 3 = Quite a bit, and 4 = Extremely). Scores ≥ 2 indicate symptom presence, and anyone providing such responses were required to complete a clinical check to ensure accurate understanding of the intended meaning and clinical relevance of the item (e.g., for item 1 measuring longing for the deceased, the clinical check states: "*This is more than just missing your loved one. It is an intense and painful desire to be with the deceased again. Is this what you felt almost every day over the past week?*"). All clinical check statements used a 'Yes' or 'No' response format. The IGQ-CC includes a question that assess if symptoms exceed what is normal in the participant's social, cultural, or religious context with response options of 'Yes', 'No', or 'I don't know'. There is also a question that assesses if the PGD symptoms caused problems in daily functioning across various domains with response options of 'Yes' or 'No'. The IGQ-CC can be found here: https://www.traumameasuresglobal.com/igq.

The psychometric properties of the IGQ have been demonstrated in multiple studies [28,29], and the internal reliability estimates for the core (α = .85), associated (α = .86), and total (α = .91) scale scores in this sample were good. Diagnostic requirements are met if (a) bereavement (in this case, including a pet) occurred more than six months ago, (b) at least one 'core' symptom and one 'associated' symptom are present (symptom presence is based on a score of 2 or higher for the initial symptom screen and a 'Yes' response to the clinical check), (c) there is evidence of functional impairment, and (d) there is a response of 'Yes' or 'I don't know' to the cultural criterion question. A strict reading of the ICD-11 would require those who answered 'I don't know' to the cultural criterion question (16.6%, *n* = 137) to be excluded from diagnostic consideration, but all prior studies have only excluded those who answered 'No' to this question. This study followed the same process as all previous studies and permitted diagnosis among those who did not know if their symptoms exceeded norms in their culture.

## Analytic plan

Descriptive statistics were used to summarise (1) what proportion of people had experienced the death of a pet, as well as all other bereavements, (2) what proportion of people who had experienced the death of a pet *and* a person identified the death of a pet as most distressing, and (3) the conditional rates of ICD-11 PGD based on each index bereavement. A

**Table 1. Sociodemographic Details for Participants (N = 975).**

| | % | n |
|---|---|---|
| *Sex* | | |
| Male | 48.5 | 473 |
| Female | 51.5 | 502 |
| *Age* | | |
| 18-24 | 11.7 | 114 |
| 25-34 | 19.6 | 191 |
| 35-44 | 18.7 | 182 |
| 45-54 | 16.8 | 164 |
| 55-64 | 13.6 | 133 |
| 65+ | 19.6 | 191 |
| Born in UK | 89.3 | 871 |
| *UK Nation* | | |
| England | 86.4 | 842 |
| Wales | 4.9 | 48 |
| Scotland | 6.8 | 66 |
| Northern Ireland | 1.9 | 19 |
| *Annual Income* | | |
| Less than £20,000 | 27.0 | 263 |
| £20,000-£39,999 | 34.5 | 336 |
| £40,000-£59,999 | 19.5 | 190 |
| £60,000-£79,999 | 10.4 | 101 |
| £80,000-£99,999 | 5.1 | 50 |
| More than £100,000 | 3.6 | 35 |
| *Highest Educational Attainment* | | |
| No Qualification | 4.0 | 39 |
| O-level/ GCSE or similar | 25.3 | 247 |
| A-level or similar | 28.5 | 278 |
| Undergraduate Degree | 28.9 | 282 |
| Postgraduate Degree | 13.2 | 129 |
| *Employment Status* | | |
| Full-time employed | 45.8 | 447 |
| Part-time employed | 17.4 | 170 |
| Unemployed, seeking work | 5.2 | 51 |
| Unemployed, not seeking work | 4.6 | 45 |
| Not working due to disability | 5.0 | 49 |
| Student | 3.1 | 30 |
| Retired | 18.8 | 183 |
| *Relationship Status* | | |
| In a committed relationship | 71.2 | 694 |
| Not in a committed relationship | 28.8 | 281 |
| *Children* | | |
| 0 | 35.4 | 345 |
| 1 | 23.4 | 228 |
| 2 | 25.4 | 248 |
| 3 | 11.2 | 109 |
| 4 or more | 4.6 | 45 |

one-sample proportion Z-test was used to determine if the conditional rate of PGD based on the death of a pet was significantly different from zero.

Relative risk and population-attributable risk percentages (PAR%) for ICD-11 PGD based on each type of bereavement were also calculated. Relative risk indicates the probability of a given event (e.g., meeting diagnostic requirements for ICD-11 PGD) for one group versus another (e.g., pet-bereaved versus not pet-bereaved). A value of 1 indicates that the risk of the outcome is the same for both groups; values > 1 indicate risk is higher in the exposed group; and values < 1 indicates risk is lower in the exposed group. The PAR% reflects the percentage of cases for a given outcome (i.e., PGD) that would be eliminated if a given event did not occur (e.g., the death of a child). It is a useful statistic because it accounts for both the strength of the effect of the assumed risk factor on the outcome variable, and the prevalence of that risk factor in the population.

Measurement invariance analysis was used to determine if PGD symptoms, as measured by the IGQ-CC, operated equivalently for those who reported their symptoms in relation to the death of a human and those who reported their symptoms in relation to the death of a pet. Prior to testing for measurement invariance, the optimal latent structure of the PGD symptoms *among all bereaved persons* was first assessed using confirmatory factor analysis (CFA). Two models were tested; a one-factor model where all items loaded onto a single factor, and a two-factor model where items 1 and 2 loaded onto a 'core symptoms' factor, and items 3–5 loaded onto an 'associated symptoms' factor. Measurement invariance testing was based on the better fitting model.

For measurement invariance, a 'configural invariance' model is first tested to determine if the same measurement model (i.e., the one- or two-factor model) holds in both groups (i.e., pet-bereavement or human-bereavement). This is tested by assessing the overall fit of the configural model to the sample data, and if the overall fit is good, configural invariance is supported. Next, a 'metric invariance' model is tested where the factor loadings are held equal across the two groups to determine if each item relates to the underlying latent factor in the same way in both groups. This is tested by comparing the fit of the metric model to the configural model, and metric invariance is assumed if there is no marked deterioration in fit for the simpler metric model. Finally, a 'scalar invariance' model is tested where the factor loadings and item intercepts are held equal across the two groups. The simpler scalar model is compared to the metric model and if there is no marked deterioration in fit, scalar invariance is supported.

All CFA and measurement invariance models were tested using robust maximum likelihood estimation in *Mplus*8 [30]. Standard guidelines for determining model fit were followed where good model fit is indicated by a non-significant chi-square ($\chi^2$) result, comparative fit index (CFI) and Tucker–Lewis index (TLI) values ≥ .90, and root mean square error of approximation (RMSEA) and standardized root mean square residual (SRMR) values ≤ .08 [31]. Alternative models were compared using the Bayesian Information Criterion (BIC) statistic where the model with the lower value is preferred. For measurement invariance testing, Chen's [32] guidelines were followed such that changes (Δ) <.010 in the CFI, <.015 in the RMSEA, and <.030 in the SRMR support invariance. Additionally, $\chi^2$ difference tests were examined whereby non-significant results support invariance.

## Results

### Research question 1

In the full sample, 84.2% ($n=821$) were bereaved when the loss of a pet was included. This figure dropped to 81.9% ($n=799$) if pet bereavement was excluded. The most common bereavement was the death of a parent (48.1%), the least common was the death of a child (6.8%), and 32.6% ($n=318$) experienced the death of a beloved pet (see Table 2 for details).

### Research question 2

Most people who had experienced the death of a pet had also experienced the death of a person (93.1%, $n=296$). When asked to identify the bereavement that caused them the most distress, 21.0% ($n=62$) of these people chose the death of a pet. Additional details are provided in Table 2.

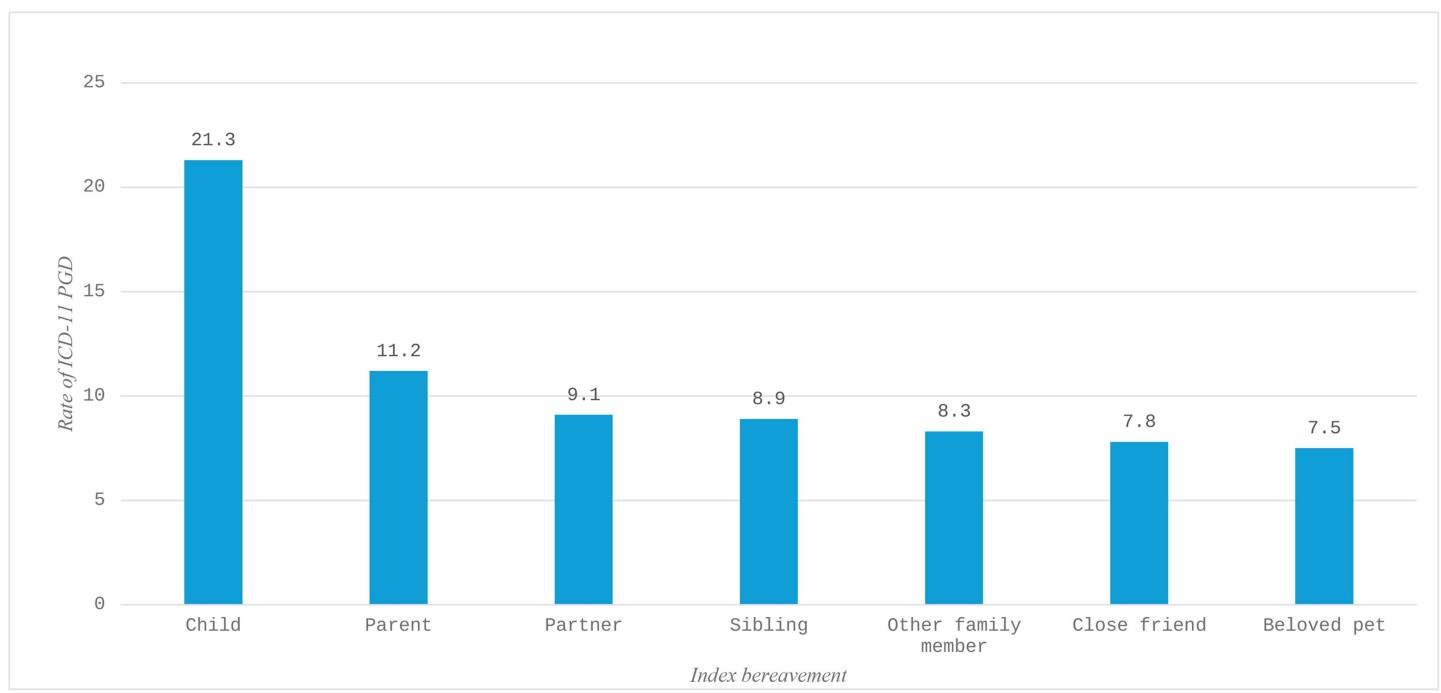

**Table 2. Proportion of UK Adults who Experienced Each Type of Bereavement and the Most Distressing Bereavement Reported by Those who lost a Pet and a Human.**

|  | Occurrence ($N=975$)[1] | | Most distressing bereavement ($n=295$)[2] | |
|---|---|---|---|---|
|  | % | n | % | n |
| Death of a parent | 48.1 | 469 | 42.0 | 124 |
| Death of a sibling | 14.2 | 138 | 4.1 | 12 |
| Death of a partner | 8.5 | 83 | 4.7 | 14 |
| Death of a child | 6.8 | 66 | 6.1 | 18 |
| Death of some other family member | 45.6 | 445 | 16.3 | 48 |
| Death of a close friend | 26.5 | 258 | 5.8 | 17 |
| Death of a beloved pet | 32.6 | 318 | 21.0 | 62 |

*Note*: [1] = Proportion of people in the entire sample who experienced each type of bereavement; [2] = Among those who experienced the death of a human and a pet (1 cases was missing), these are the proportions identifying their most distressing bereavement.

## Research question 3

Among all participants, 8.6% ($n=84$) met diagnostic requirements for ICD-11 PGD. Conditional rates of probable PGD based on each index bereavement are presented in Fig 1. The highest conditional rate was for the death of a child (21.3%), and the lowest was for the death of a pet (7.5%) but this figure was similar to many types of human bereavement. The conditional rate of ICD-11 PGD following the death of a pet was significantly different from zero ($Z=7.60$, $p<.001$).

**Fig 1. Conditional rates of ICD-11 PGD in the sample based on most distressing bereavement (N = 975).**

## Research question 4

The relative risk and PAR% values for ICD-11 PGD based on each type of bereavement are presented in Table 3. Relative risk values ranged from 0.87 (death of a close friend) to 2.08 (death of a child), and only those values associated with the death of a child and the death of a partner were statistically significant ($p < .05$). The relative risk value for the death of a pet was 1.27. The largest PAR% was associated with the death of the parent (12.8%), followed by the death of a pet (8.1%).

## Research question 5

Among all bereaved persons, the one-factor model had acceptable fit to the sample data ($x^2$ (df = 5, $n$ = 820) = 32.576, $p < .001$; CFI = .980; TLI = .959; RMSEA (90% CI) =.082 (.057,.110); SRMR = .021; BIC = 10,985), whereas the two-factor model had extremely close fit to the data ($x^2$ (df = 4, $n$ = 820) = 6.672, $p$ = .154; CFI = .998; TLI = .995; RMSEA (90% CI) =.029 (.000,.065); SRMR = .009; BIC = 10,943). Thus, the two-factor model was selected as the optimal representation of the latent structure of the PGD symptoms. All standardized factor loadings were positive, significant ($p < .001$), and strong (ranging from.81 to.88), and the two factors were strongly correlated (r = .92, $p < .001$).

Model fit results for the configural, metric, and scalar invariance models are presented in Table 4. The configural model fit the sample data closely, indicating that the two-factor model represented the latent structure of the PGD symptoms equally well for those who reported their symptoms in relation to death of a pet and in relation to the death of a person. The metric invariance model (equal factor loadings) fit the data closely with no deterioration in fit relative to the configural model ($\Delta x^2$ (3) = 0.93, $p$ = .818). The scalar invariance model (equal factor loadings and equal item intercepts) also fit the data closely, with no significant difference in fit compared to the metric model ($\Delta x^2$ (3) = 5.71, $p$ = .127).

**Table 3. Relative Risk (RR) and Population Attributable Risk Percentage (PAR%) Values for ICD-11 PGD Based on Each Type of Bereavement.**

|  | RR (95% CI) | PAR% |
|---|---|---|
| Death of a parent | 1.31 (0.87, 1.97) | 12.8% |
| Death of a sibling | 1.21 (0.70, 2.09) | 2.9% |
| Death of a partner | **1.79 (1.02, 3.16)** | 6.3% |
| Death of a child | **2.08 (1.16, 3.72)** | 6.8% |
| Death of some other family member | 0.94 (0.62, 1.42) | −2.9% |
| Death of a close friend | 0.87 (0.54, 1.41) | −3.6% |
| Death of a beloved pet | 1.27 (0.84, 1.93) | 8.1% |

Note: 95% CI = 95% confidence interval for the risk ratio value; statistically significant ($p < .05$) risk ratio in bold.

**Table 4. Measurement Invariance Tests Results for the International Grief Questionnaire.**

|  | $x^2$ | df | p | CFI | TLI | RMSEA (90% CI) | SRMR | BIC |
|---|---|---|---|---|---|---|---|---|
| Configural | 14.269 | 8 | .075 | .996 | .989 | .044 (.000,.080) | .012 | 10986 |
| Metric | 16.497 | 11 | .124 | .996 | .993 | .035 (.000,.068) | .013 | 10967 |
| Scalar | 21.800 | 14 | .083 | .995 | .992 | .037 (.000 −.066) | .015 | 10952 |

Note: Estimator = MLR; χ2 = Chi-square Goodness of Fit statistic; df = degrees of freedom; p = Statistical significance; CFI = Comparative Fit Index; TLI = Tucker Lewis Index; RMSEA (90% CI) = Root-Mean-Square Error of Approximation with 90% confidence intervals; SRMR = Standardized Root-Mean Square Residual; BIC = Bayesian Information Criterion.

## Discussion

In 2024, 51% of UK adults owned a pet [33]; a figure slightly lower than in the United States where 63% of adults own a pet [20]. The relatively short lifespans of companion animals like cats, dogs, and rabbits (the most commonly owned pets) means that most pet owners will live long enough to experience the death of one or more of their pets. Many pet owners experience intense grief following the death of their pet [21–23], and many also report feelings of shame, embarrassment, and isolation as a result of expressing their grief for their deceased pet [34]. Negative social reactions to displays of grief have been labelled 'disenfranchised grief' [35], and this phenomenon has been identified as being particularly relevant to those mourning a pet [36]. If people can develop clinically significant levels of grief following the death of a pet, then it is essential that this is recognised in the scientific literature so that mental health professionals can communicate with the public in an appropriate and accurate manner, and people who need, and desire, clinical care are afforded the opportunity to access it. It was for this reason that the current study was performed.

Results indicated that about a third of UK adults had experienced the death of a beloved pet, and almost all had also experienced the death of a person who they were close to. Recognising that multiple bereavements are common, participants were asked to nominate their most distressing loss, and just over one-in-five people who had lost a beloved pet and a person they were close to stated that the loss of their pet was most distressing. Thus, for many people the loss of a pet is perceived to be worse than the loss of a person. This is consistent with the existing evidence that people form close emotional bonds with their pets [e.g., 15, 17, 20].

Following the death of a pet, 7.5% met diagnostic requirements for ICD-11 PGD. This figure was similar to conditional rates for the death of a close friend (7.8%), a family member such as a grandparent, cousin, aunt/uncle (8.3%), a sibling (8.9%), and even a partner (9.1%). Only the death of a parent (11.2%), and in particular, the death of a child (21.3%) were markedly higher. What should be made of these findings? ICD-11 and DSM-5-TR do not allow for PGD to be diagnosed following the death of a pet, but these results demonstrate that people can experience clinically relevant levels of grief after the death of a pet, and at rates that are comparable to human losses that are considered 'legitimate' risk factors for PGD. Replication of these results is, of course, essential. However, if confirmed, proponents of the current bereavement criterion would be required to hold the view that an individual can satisfy all symptom and impairment requirements for PGD yet be ineligible for diagnosis solely because the deceased was not a member of the *homo sapiens sapiens* species. From both psychological and evolutionary standpoints, this would be an extraordinarily difficult position to defend.

The relative risk and PAR% values for ICD-11 PGD associated with each type of bereavement were calculated to quantify the impact of pet bereavement relative to other types of human bereavements. Notably, only two types of losses were associated with statistically significant relative risk values. The death of a child and the death of a partner were both associated with a two-fold increased risk of PGD. These results are consistent with existing evidence that the loss of a child and the loss of a spouse are most strongly associated with meeting diagnostic requirements for PGD [11–14]. Participants who lost a pet were 27% more likely to meet diagnostic requirements for PGD relative to those who had not lost a pet. What was noteworthy about this result was that it was of a similar magnitude to that observed for the death of a parent (31%) and the death of a sibling (21%), and stronger than that observed for the death of a close friend and the death of some other family member. The PAR% was 8.1% for the death of a pet, meaning that pet bereavement accounts for approximately 1-in-12 PGD cases in the population, second only to parental bereavement. Notably, the PAR% value for pet loss was higher than the death of a partner and the death of child. Even though these two types of losses carry more risk of PGD in absolute terms, pet bereavement contributes more cases of PGD in the population given its greater frequency of occurrence.

The final set of analyses examined if PGD symptoms were measured in the same way between people who reported their symptoms in response to a pet loss and in response to a human loss. If, as stipulated by the ICD-11 and DSM-5-TR diagnostic formulations, there is something unique about grief that is experienced following a human bereavement, then symptoms of PGD should manifest differently between these two groups of bereaved people. One the other hand, if there

is no difference in how grief manifests in response to human and non-human bereavements, one would expect to find that the symptoms manifest in the same way. The findings of the measurement invariance analyses were clear that PGD symptoms operated identically for those who experienced the death of a pet to those who experienced the death of a human. These results demonstrate that there is nothing unique or special about how PGD symptoms are experienced in relation to a human bereavement.

These results should be interpreted considering several limitations. First, the non-probability nature of the sampling procedure, and the online recruitment method, limits the generalizability of the findings to the entire UK population. Moreover, whether these findings generalize to culturally distinct populations with different norms regarding pet ownership is unknown. Second, cause of death can influence risk of PGD [11], and pet owners often face unique challenges such as deciding whether or not to euthanise their pet, and this can influence the grieving process [37,38]. Future studies should explore whether cause of death moderates the grieving process following pet and human bereavement. Third, the IGQ-CC was used to measure PGD symptoms in this study, and psychometric support for this scale has only ever been obtained in samples of human bereaved individuals, prior to this study.

## Conclusions

These findings provide consistent and compelling evidence that people can experience clinically relevant levels of PGD following the death of a pet. Considered in light of evidence that people view grief related to the death of a pet as less legitimate than grief related to the death of a person [24], and that many people grieving the loss of their pet feel embarrassed and isolated as a result [34], the decision to exclude pet loss from the bereavement criterion for PGD can be viewed as not only scientifically misguided, but also as callous.

## Author contributions

**Conceptualization:** Philip Hyland.

**Data curation:** Philip Hyland.

**Formal analysis:** Philip Hyland.

**Funding acquisition:** Philip Hyland.

**Investigation:** Philip Hyland.

**Methodology:** Philip Hyland.

**Project administration:** Philip Hyland.

**Resources:** Philip Hyland.

**Software:** Philip Hyland.

**Writing – original draft:** Philip Hyland.

**Writing – review & editing:** Philip Hyland.

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
