## [Decision Letter · Decision Letter 0]

5 Nov 2025

Dear Dr. Hyland,

Thank you for submitting your manuscript to PLOS ONE. After careful consideration, we feel that it has merit but does not fully meet PLOS ONE’s publication criteria as it currently stands. Therefore, we invite you to submit a revised version of the manuscript that addresses the points raised during the review process.

We look forward to receiving your revised manuscript.

Kind regards,

Jose A. Calvache, MD, MSc, PhD

Academic Editor

PLOS ONE

Journal Requirements:

2. Please note that your Data Availability Statement is currently missing the repository name. If your manuscript is accepted for publication, you will be asked to provide these details on a very short timeline. We therefore suggest that you provide this information now, though we will not hold up the peer review process if you are unable.

**Additional Editor Comments:**

Dear authots,

The study employed a cross-sectional online survey design, which is appropriate for descriptive and comparative purposes; however, such a design inherently limits the ability to draw causal inferences or make definitive diagnostic claims. The use of the International Grief Questionnaire with “clinical checks” (IGQ-CC) was methodologically sound and supported by previous literature demonstrating strong psychometric properties. Nonetheless, its cross-cultural validity has not been specifically established for cases of pet loss, which may limit the generalizability of the findings. The diagnosis of Prolonged Grief Disorder (PGD) was based solely on self-reported measures without clinical confirmation or structured interviews, which could lead to over- or underestimation of actual cases. Despite this limitation, the measurement of the construct remains psychometrically supported. Thanks for the submission.

The statistical analyses were clearly described and appropriate to the study’s objectives; however, the absence of multivariate models controlling for demographic variables—such as age, gender, education, type of pet, or time since the loss—restricts the interpretability of the reported associations and precludes conclusions about potential confounding effects. The data presented are coherent and support the central finding that symptoms of PGD may also manifest following the death of a pet. Nonetheless, the study’s conclusions should remain interpretive rather than prescriptive, as the evidence—derived from a single cross-sectional, self-report design—does not provide a sufficient basis for causal claims or for advocating diagnostic manual revisions. i suggest to avoid any causal claims and present your study as descritive modelling with an interesting and provocative result.

Reviewers' comments:

Reviewer's Responses to Questions

**Comments to the Author**

1. Is the manuscript technically sound, and do the data support the conclusions?

Reviewer #1: Yes

Reviewer #2: Yes

2. Has the statistical analysis been performed appropriately and rigorously?

Reviewer #1: Yes

Reviewer #2: Yes

3. Have the authors made all data underlying the findings in their manuscript fully available?

Reviewer #1: Yes

Reviewer #2: Yes

4. Is the manuscript presented in an intelligible fashion and written in standard English?

Reviewer #1: Yes

Reviewer #2: Yes

Reviewer #1: CONSIDERATIONS:

The subject is treated in a complete way; the article is solid.

It has a scientific interest, complements the existing information in literature, the original contributions are well achieved.

The results are well elaborated, the discussion and conclusions are strong, their contribution is important to the existing literature.

There is overall coherence in the development of the article.

The references are written in Vancouver style; however, I did not find the referencing in the text, the bibliography is not listed.

The wording is clear and precise.

The figures and tables are understandable, I suggest reworking figure 1, it looks pixelated.

The methodology is well constructed, a remark is made in the "notes" section.

The discussion is good and well structured, clinically relevant issues are discussed.

It is a novel and necessary issue to generate an update of the definition of PGD in future consensus.

NOTES:

Page 8: Scores ≥ to… (¿?), incomplete information.

The study uses the cultural criterion of PGD (symptoms that exceed what is normal in the participant's social/cultural/religious context). It is noted in a footnote that a strict reading of ICD-11 could exclude cases that answered, "I don't know" (16.6%, n=137), but that previous studies followed that only excluded those that answered "No". The rationale for this decision (including n=137) should be moved to the methods section and discussed further as a limitation or a key methodological decision. The impact of strict exclusion of these cases on PGD rates should be briefly considered in the Discussion.

I suggest mentioning as a limitation the fact that the selection of participants from online research panels (Qualtrics) could bias the results (for example, towards those with greater digital access or certain demographic trends not covered by quotas).

Reviewer #2: The research utilizes a cross-sectional approach through an online survey, which is suitable for both descriptive and comparative aims. Nevertheless, this kind of design does not permit causal conclusions or conclusive diagnostic claims.

Concerning the instrument utilized, the use of the International Grief Questionnaire with “clinical checks” is suitable and backed by prior literature highlighting its strong psychometric characteristics. However, it is crucial to highlight that the cross-cultural validity of the IGQ-CC was not confirmed for the particular situation of pet loss, which could restrict the applicability of the results.

The diagnosis of Prolonged Grief Disorder depended solely on self-reported assessments, lacking clinical validation or organized interviews. This methodological constraint might result in either an overcount or an undercount of real cases. Nonetheless, the measurement of the construct is regarded as reliable and backed by previous psychometric findings.

The statistical methods are clearly outlined and deemed suitable for the objectives specified. Nevertheless, multivariate analyses were not performed to account for possible demographic covariates—like age, gender, education level, type of pet, or time since the loss—thus restricting the interpretation of relative risk estimates and the accuracy of the reported associations.

The data provided are coherent and back the study's key findings, especially the point that PGD symptoms can also arise after the passing of a pet. However, the normative or prescriptive conclusions—like the recommendation to revise international diagnostic manuals—lack adequate empirical backing, particularly since the results stem from a solitary cross-sectional self-report study.

The manuscript ultimately details ethical approval and quality control protocols implemented during data collection. It further shows that the data can be accessed publicly on OSF, and the authors state that there are no conflicts of interest or external funding, thereby adhering to the journal's ethical and transparency requirements.

**Do you want your identity to be public for this peer review?** For information about this choice, including consent withdrawal, please see our Privacy Policy

Reviewer #1: No

Reviewer #2: No

---

## [Author Response · Author response to Decision Letter 1]

14 Nov 2025

Editor comments:

Reply: The documents on style requirements have been reviewed, and every effort have been made to ensure that the manuscript is presented in-line with the journal requirements.

2. Please note that your Data Availability Statement is currently missing the repository name. If your manuscript is accepted for publication, you will be asked to provide these details on a very short timeline. We therefore suggest that you provide this information now, though we will not hold up the peer review process if you are unable.

Reply: The dataset has been uploaded to a publicly accessible OSF page where it can be downloaded. We have added the following to the manuscript in the methods section: “The dataset used for this study is publicly available on the Open Science Framework (https://osf.io/hymd4/overview).”

Reply: As above, the dataset has been uploaded to a publicly accessible OSF webpage and a link to the dataset is included in the manuscript.

Reply: Noted.

Reply: Noted.

Additional Editor Comments:

Dear authors,

The study employed a cross-sectional online survey design, which is appropriate for descriptive and comparative purposes; however, such a design inherently limits the ability to draw causal inferences or make definitive diagnostic claims. The use of the International Grief Questionnaire with “clinical checks” (IGQ-CC) was methodologically sound and supported by previous literature demonstrating strong psychometric properties. Nonetheless, its cross-cultural validity has not been specifically established for cases of pet loss, which may limit the generalizability of the findings. The diagnosis of Prolonged Grief Disorder (PGD) was based solely on self-reported measures without clinical confirmation or structured interviews, which could lead to over- or underestimation of actual cases. Despite this limitation, the measurement of the construct remains psychometrically supported. Thanks for the submission.

The statistical analyses were clearly described and appropriate to the study’s objectives; however, the absence of multivariate models controlling for demographic variables—such as age, gender, education, type of pet, or time since the loss—restricts the interpretability of the reported associations and precludes conclusions about potential confounding effects. The data presented are coherent and support the central finding that symptoms of PGD may also manifest following the death of a pet. Nonetheless, the study’s conclusions should remain interpretive rather than prescriptive, as the evidence—derived from a single cross-sectional, self-report design—does not provide a sufficient basis for causal claims or for advocating diagnostic manual revisions. I suggest to avoid any causal claims and present your study as descriptive modelling with an interesting and provocative result.

Reply: Duly noted, and great care has been taken in the revision phase to ensure that all language used, and conclusions drawn, fairly and accurately reflect the methodological design of the study.

Reviewer #1: CONSIDERATIONS:

The subject is treated in a complete way; the article is solid.

Reply: Thank you.

It has a scientific interest, complements the existing information in literature, the original contributions are well achieved.

Reply: Thank you.

The results are well elaborated, the discussion and conclusions are strong, their contribution is important to the existing literature.

Reply: Thank you.

There is overall coherence in the development of the article.

Reply: Thank you.

The references are written in Vancouver style; however, I did not find the referencing in the text, the bibliography is not listed.

Reply: The references have been checked and now match the journal requirements.

The wording is clear and precise.

Reply: Thank you.

The figures and tables are understandable, I suggest reworking figure 1, it looks pixelated.

Reply: Figure 1 on my end looks clear and detailed. I am happy to work with the editor to ensure that the figure resolution is of the highest possible standard.

The methodology is well constructed, a remark is made in the "notes" section.

Reply: Thank you. The footnote has been moved into the text of the methods based on the request of reviewer 2.

The discussion is good and well structured, clinically relevant issues are discussed.

Reply: Thank you.

It is a novel and necessary issue to generate an update of the definition of PGD in future consensus.

Reply: Thank you.

Page 8: Scores ≥ to… (¿?), incomplete information.

Reply: This is now updated to note that scores >= 2 indicate symptom presence.

The study uses the cultural criterion of PGD (symptoms that exceed what is normal in the participant's social/cultural/religious context). It is noted in a footnote that a strict reading of ICD-11 could exclude cases that answered, "I don't know" (16.6%, n=137), but that previous studies followed that only excluded those that answered "No". The rationale for this decision (including n=137) should be moved to the methods section and discussed further as a limitation or a key methodological decision. The impact of strict exclusion of these cases on PGD rates should be briefly considered in the Discussion.

Reply: Noted, this change has been made to the manuscript.

I suggest mentioning as a limitation the fact that the selection of participants from online research panels (Qualtrics) could bias the results (for example, towards those with greater digital access or certain demographic trends not covered by quotas).

Reply: This has been added to the limitations, specifically to the first limitation where the limits of generalizability are discussed.

Reviewer #2 comments:

The research utilizes a cross-sectional approach through an online survey, which is suitable for both descriptive and comparative aims. Nevertheless, this kind of design does not permit causal conclusions or conclusive diagnostic claims.

Concerning the instrument utilized, the use of the International Grief Questionnaire with “clinical checks” is suitable and backed by prior literature highlighting its strong psychometric characteristics. However, it is crucial to highlight that the cross-cultural validity of the IGQ-CC was not confirmed for the particular situation of pet loss, which could restrict the applicability of the results.

Reply: Noted, and a sentence has been added to the limitations section: “Third, the IGQ-CC was used to measure PGD symptoms in this study, and psychometric support for this scale has only ever been obtained in samples of human bereaved individuals, prior to this study.”

The diagnosis of Prolonged Grief Disorder depended solely on self-reported assessments, lacking clinical validation or organized interviews. This methodological constraint might result in either an overcount or an undercount of real cases. Nonetheless, the measurement of the construct is regarded as reliable and backed by previous psychometric findings.

Reply: Thank you.

The statistical methods are clearly outlined and deemed suitable for the objectives specified. Nevertheless, multivariate analyses were not performed to account for possible demographic covariates—like age, gender, education level, type of pet, or time since the loss—thus restricting the interpretation of relative risk estimates and the accuracy of the reported associations.

Reply: I see the point, but the research questions under investigation did not necessitate covariate adjustment.

The data provided are coherent and back the study's key findings, especially the point that PGD symptoms can also arise after the passing of a pet. However, the normative or prescriptive conclusions—like the recommendation to revise international diagnostic manuals—lack adequate empirical backing, particularly since the results stem from a solitary cross-sectional self-report study.

Reply: Good point. The rather extreme statement at the end of the discussion that the diagnostic requirements should changed has been deleted.

The manuscript ultimately details ethical approval and quality control protocols implemented during data collection. It further shows that the data can be accessed publicly on OSF, and the authors state that there are no conflicts of interest or external funding, thereby adhering to the journal's ethical and transparency requirements.

Reply: Thank you.

---

## [Editor Report · Decision Letter 1]

2 Dec 2025

No pets allowed: Evidence that prolonged grief disorder can occur following the death of a pet

PONE-D-25-52202R1

Dear Dr. Hyland,

We’re pleased to inform you that your manuscript has been judged scientifically suitable for publication and will be formally accepted for publication once it meets all outstanding technical requirements.

Kind regards,

Jose A. Calvache, MD, MSc, PhD

Academic Editor

PLOS ONE

Additional Editor Comments (optional):

We would like to thank the authors for their thorough revisions and for providing comprehensive answers to all of the reviewers' questions and comments. The article now meets the high standards of the journal and is accepted for publication. Congratulations!
---

## [Editor Report · Acceptance letter]

PONE-D-25-52202R1

PLOS One

Dear Dr. Hyland,

I'm pleased to inform you that your manuscript has been deemed suitable for publication in PLOS One. Congratulations! Your manuscript is now being handed over to our production team.

Kind regards,

on behalf of

Dr. Jose A. Calvache

Academic Editor

PLOS One